# Importance of the Microenvironment and Mechanosensing in Adipose Tissue Biology

**DOI:** 10.3390/cells11152310

**Published:** 2022-07-27

**Authors:** Simon Lecoutre, Mélanie Lambert, Krzysztof Drygalski, Isabelle Dugail, Salwan Maqdasy, Mathieu Hautefeuille, Karine Clément

**Affiliations:** 1Nutrition and Obesities: Systemic Approaches Research Group (Nutri-Omics), Sorbonne Université, INSERM, F-75013 Paris, France; 23simon.lec@gmail.com (S.L.); drygalskikrzysztof@gmail.com (K.D.); isabelle.dugail@inserm.fr (I.D.); 2Labex Inflamex, Université Sorbonne Paris Nord, INSERM, F-93000 Bobigny, France; melanie.lambert@univ-paris13.fr; 3Department of Medicine (H7), Karolinska Institutet Hospital, C2-94, 14186 Stockholm, Sweden; salwan.maqdasy@ki.se; 4Laboratoire de Biologie du Développement (UMR 7622), IBPS, Sorbonne Université, F-75005 Paris, France; mathieu.hautefeuille@sorbonne-universite.fr; 5Assistance Publique Hôpitaux de Paris, Nutrition Department, CRNH Ile-de-France, Pitié-Salpêtrière Hospital, F-75013 Paris, France

**Keywords:** adipose tissue, obesity, mechanobiology

## Abstract

The expansion of adipose tissue is an adaptive mechanism that increases nutrient buffering capacity in response to an overall positive energy balance. Over the course of expansion, the adipose microenvironment undergoes continual remodeling to maintain its structural and functional integrity. However, in the long run, adipose tissue remodeling, typically characterized by adipocyte hypertrophy, immune cells infiltration, fibrosis and changes in vascular architecture, generates mechanical stress on adipose cells. This mechanical stimulus is then transduced into a biochemical signal that alters adipose function through mechanotransduction. In this review, we describe the physical changes occurring during adipose tissue remodeling, and how they regulate adipose cell physiology and promote obesity-associated dysfunction in adipose tissue.

## 1. Introduction

The dramatic rising incidence of obesity has increased the urgent need to understand underlying mechanisms of obesity and its link with numerous co-morbidities. Obesity is now one of the greatest public health issues affecting ~13% of adults worldwide [1]. Obesity leads to the development of complications such as type-2 diabetes, non-alcoholic fatty liver disease, cardiovascular diseases, neurodegenerative disorders, greater predisposition to serious infections (including COVID-19) and cancers [2,3,4].

Adipose tissue (AT) expansion and its altered function is the cornerstone of obesity-related complication development. AT is a highly dynamic organ that modifies its size, cellular composition, and function, upon nutritional and hormonal challenges (Appendix A). In obesity, white adipose tissue (WAT) can expand by 10 times through hypertrophy (enlarged adipocyte size) and hyperplasia (increased adipocyte number), whereas upon fasting or cold exposure, WAT shrinks by lipolysis to supply fatty acids to peripheral organs [5,6]. In addition to their storage capacity, adipocytes are able to sense the energy state and secrete factors called adipokines (including lipokines) to coordinate, and to regulate, whole body metabolism [7]. In obesity, adipocytes are exposed to different forms of stresses such as hypoxic, oxidative, metabolic, inflammatory and mechanical stressors [8,9,10]. However, the precise triggers for adipocyte dysfunction and their coordinated mode of action still remain uncertain. Nevertheless, there is little doubt that adipocyte hypertrophy is a major factor of adipocyte dysfunction, suggesting that this stimulus requires careful attention. It is indeed known that the fat cell size correlates with the severity of insulin resistance and type 2 diabetes mellitus as well as altered responses to weight loss [11,12,13,14,15].

AT expansion is a complex biological process consisting of major coordinated and intertwined events: (i) increase in the size of adipocytes triggered by excessive lipogenesis and triglyceride storage, (ii) infiltration of immune cells in the parenchyma and around adipocytes, (iii) remodeling of the extracellular matrix (ECM), (iv) formation of new adipocytes by adipogenesis from progenitor cells (Appendix B), and (v) angiogenesis. Adipocytes adapt to a dynamic niche as well as a microenvironment invaded by foreign cells in its stromal vascular fraction (SVF). However, the adaptation potential is limited in time and space, whereby the inability of WAT to adequately expand to meet the energy storage demands results in AT dysfunction, ectopic lipid deposition, and insulin resistance [16]. Whereas ECM is recognized as a critical component of the AT microenvironment [9,10], the resulting altered tissue mechanics in obesity need to be better studied, since this phenomenon can be a potential driver of AT dysfunction. This also requires deeper understanding of adipocyte mechanotransduction. Indeed, during AT expansion, tightly controlled tissue homeostasis is lost, and dysregulation of the ECM may alter the mechano-reciprocity between adipocytes, SVF cells and the ECM to create an iatrogenic feedback loop.

In this review, we discuss the emerging role of mechanosensitive pathways in regulating AT function and the potential repercussions of altered adipocyte mechanical microenvironment in the initiation of AT-related diseases.

## 2. Major Alterations of Expanded Adipose Tissue

### 2.1. Hypertrophy vs. Hyperplasia

In adulthood, AT size is fairly constant under homeostatic conditions, but it is highly sensitive to dietary challenges (e.g., high caloric diet). In a given fat depot, the number of adipocytes is determined early in life and is mostly stable through adulthood [17,18,19,20]. Since mature adipocytes are post-mitotic cells, new cells are derived from adipose stem cells (ASC) located in the SVF (Appendix B) [10,17,21]. In young adult mice, the rate of adipogenesis has been estimated to 10–15% per month [19,20], and retrospective human studies have estimated the turnover rate to 10% per year [18]. However, the turnover rate of adipocytes gradually declines with age due to defective regenerative ability of the ASCs and the senescence of the microenvironment [20,22].

In response to caloric excess, mature adipocytes possess impressive hypertrophic potential, being able to increase in size to several hundreds of micrometers in diameter [23]. Moreover, fat mass might also expand by hyperplasia [19,20,24,25,26]. For instance, it has been shown that visceral WAT expansion in males occurs through both adipocyte hyperplasia and hypertrophy in both mice [24,27] and humans [28]. In male mice, subcutaneous WAT does not exhibit relevant hyperplasia in response to obesogenic stimuli [24,25,27,29] while in female mice, high-fat diet (HFD) induces hyperplasia in both visceral and subcutaneous WAT [27]. Thus, hyperplasia is induced in a sex hormone-dependent manner in WAT depots [27]. In 1956, Jean Vague was the first to show the importance of regional WAT distribution in relation to various metabolic diseases [30]. He observed that individuals with obesity who preferentially expand visceral WAT (i.e., “apple-shaped obesity”) are at greater risk for metabolic disorders than those who accumulate subcutaneous WAT (i.e., “pear-shaped obesity”) [30]. Recent studies suggest that hypertrophy in the subcutaneous WAT is linked to insulin resistance while visceral WAT hypertrophy also correlates with dyslipidemia [13]. Moreover, while adipocyte number in subcutaneous WAT protects against metabolic complications, an increase in the number of fat cells in visceral depots does not associate with metabolic state [31].

### 2.2. Inflammation and Fibrosis

In subjects with obesity, WAT expansion is characterized by local inflammation followed by ECM deposition. AT secretome is usually characterized by lower adiponectin and increased levels of leptin, free fatty acids, tumor necrosis factor α (TNFα), interleukin-6 (IL-6), IL-8, monocyte chemoattractant protein-1 (MCP1), and acute-phase serum amyloid A proteins [7,32]. Some inflammatory mediators are produced by hypertrophic adipocytes [33] but tissue-resident immune cells are also major contributors to the AT secretome in obesity [8,34]. Immune cells indeed dramatically increase in abundance during WAT expansion [35,36]. AT-derived immune cells display distinct phenotypes from their circulating counterparts [37,38]. AT is indeed enriched with an important diversity of immune cells including macrophages forming “crown-like” structures around hypertrophic adipocytes [39,40], regulatory T and B cells [41,42,43], lymphocytes T gamma delta and natural killer T (NKT) cells that drive thermogenesis [44,45,46], memory T cells [47], cytotoxic innate lymphoid cells [48], dendritic cells [39,49], neutrophils [50], as well as, ILC2 cells [51,52,53,54,55]. This network of immune cells is tailored to support AT homeostasis, enabling it to adapt to environmental and nutritional factors [8,56,57,58]. In obesity, the immune system of AT is dysregulated, driving sterile inflammation with altered remodeling that can progress towards ECM deposition and fibrosis accumulation [59,60] (Appendix C). Moreover, basal membrane components such as collagen IV, nidogen and proteoglycans are increased in obese AT and associate with insulin-resistance markers in subjects with obesity [61,62].

Progression towards pathological AT remodeling is often accompanied by decreased ECM flexibility possibly due to pericellular collagen deposition and enhanced ECM crosslinking by lysyl oxidase (LOX) enzymes [63]. This promotes the formation of collagen bundles that stiffen the tissue and constrain AT expansion [64]. The production of ECM proteins is also regulated by a variety of AT cells including adipocyte progenitors, adipocytes, fibroblasts, and myofibroblasts [9,65]. AT fibrosis is an aggravating factor for metabolic condition [10,66,67]. For instance, *Col6a1* knockout mouse model [66] that results in a complete loss of collagen VI fibers within the ECM in AT is accompanied by uncontrolled enlargement of the adipocytes in obesogenic condition. Although massively obese, these transgenic mice exhibit an improved metabolic and inflammatory profile [66]. In subjects with obesity, collagen VI expression levels are linked to WAT remodeling and insulin resistance [68,69]. In accordance, it was shown that subjects with obesity and insulin-resistance have increased subcutaneous and omental AT fibrosis, compared with insulin-sensitive obese subjects [61,70]. Notably, a histological study reported that pericellular fibrosis in omental AT was not associated with significant adipocyte hypertrophy [61]. It was hypothesized that pericellular fibrosis in omental AT restricts adipocyte hypertrophy which may, in turn, preserve adipocyte function and systemic metabolism [71]. However, omental mature adipocytes from insulin-resistant obese individuals display features of enhanced inflammation, oxidative stress and endoplasmic reticulum stress, compared with normoglycemia obese individuals [72].

Altogether, the AT microenvironment is a complex system composed of a myriad of cells including adipocytes, endothelial and immune cells, ASC, and fibroblasts. These cells interact with their surroundings via both mono- and heterotypic cell–cell, or cell–ECM interaction [73]. AT cells receive and integrate information from their microenvironment impacting on their behavior, including the control of expansion/shrinking and remodeling upon nutritional or environmental stimuli.

## 3. Cell–Cell Cross-Talk in Adipose Tissue

Cells of multicellular organisms need to communicate with each other to maintain tissue function and homeostasis. Intercellular crosstalk often involves direct physical cell–cell contact between adjacent cells via specific cell surface receptors and/or via soluble factors such as cytokines, chemokines, growth factors, neurotransmitters, and extracellular vesicles.

### 3.1. Connexins and Neuronal Fibers

Direct cell–cell contacts between adipocytes are mediated by connexin 43 gap junctions [74,75,76]. A gap junction is composed of two hemichannels (e.g., connexons), each of which is a hexamer of connexin subunits forming an aqueous channel with a molecular weight cut-off of approximately 1 kDa [77,78]. The connexin gap is large enough to enable the passage of ions, cAMP, and other small molecules and metabolites [79]. Scherer and colleagues demonstrated that connexin 43 facilitates the propagation of sympathetic neuronal signals across adipocytes, to enable beiging [76]. Sympathetic nerve fibers release noradrenaline implicated in mediating adipocyte lipolysis and thermogenesis [80]. However, as WAT is not well innervated compared with brown adipose tissue (BAT) [81], the gap junctions are critical to maximize the physical interactions between white adipocytes allowing an effective propagation of neuronal signals in response to stimuli. Moreover, sympathetic fibers show close apposition to >90% of adipocytes [82] and can form neuroadipocyte junctions in mouse WAT [83]. Surgical or pharmacological destruction of sympathetic fibers inhibits the leptin-stimulated lipolytic response of WAT [83]. However, the mechanism of neural regulation in AT remains elusive. Adipocytes produce neurotrophic factors including nerve growth factor [84], S-100 protein β-chain [85,86] and neuregulin 4 [87] which control proliferation, survival, migration and differentiation of neurons. Cold exposure in mice or fasting increase the expression of these neurotrophic factors that promote densification of the sympathetic axonal tree in AT [81,82,88,89]. In obese condition, AT sympathetic innervation is significantly reduced as shown in *Ob*/*Ob* compared with lean mice and subcutaneous WAT from subjects with obesity [90]. Reduced sympathetic innervation is associated with attenuated catecholamine-induced lipolysis, promoting WAT expansion and reducing the ability to withstand starvation and thermogenesis [91]. Immunohistochemical analysis of selective neuron markers, such as tyrosine hydroxylase, revealed sympathetic nerve fibers in close anatomical association with the vasculature in AT [80,92].

### 3.2. Vasculature

The vasculature supplies the AT microenvironment with the necessary oxygen, nutrients, hormones, and growth factors, and removes metabolic waste products. Vasculature is known to play an important role in the regulation of triglyceride storage. Lipoprotein lipase (LPL), which is mainly synthesized by the adipocyte, is an extracellular enzyme that acts on the endothelial surface and hydrolyzes the triglycerides of chylomicrons to fatty acids [93,94], that are then up taken by FAT/CD36 (fatty acid translocase), the principal adipocyte fatty acid transporter [95,96]. The vasculature contributes to the maintenance of adipocyte turnover [97]. Several populations of ASCs have been identified in the mural compartment of vascular structures [97,98,99,100]. Furthermore, a study using irradiation followed by bone marrow transplantation suggests that over 85% of macrophages are recruited from blood monocytes into AT in a CSF1-dependent manner [35]. Interestingly, sympathetic activation promotes angiogenesis via upregulation of VEGF-A expression in rat BAT [101,102]. Adipocytes secrete VEGF which acts on the VEGFR2 receptor expressed on the endothelium to stimulate angiogenesis [103]. Induction of VEGF expression in adipocytes increases adipose vasculature, improves AT function and reduces insulin resistance and glucose intolerance in HFD fed mice [104,105]. Whether these results obtained in mouse models can be translated to human is still under debate [106].

### 3.3. Immune Cells

Resident AT immune cells are necessary for adipocyte biological function. In the lean condition, the immune cells are mostly anti-inflammatory, comprising alternatively activated (M2) macrophages which maintain their polarization through type 2 innate lymphoid cells (ILC2), T regulatory cells (Tregs), invariant natural killer T cells (iNKT cells), natural killer cells, and eosinophils [106]. Macrophages are responsible for many housekeeping processes, such as the removal of dead cells, ECM remodeling, regulation of catecholamine availability, modulation of angiogenesis and differentiation of adipocyte precursors [83,107,108]. At an early stage, the inflammatory reactions are necessary for AT expansion. However, in obesity, the adipose immune system shifts toward a chronic pro-inflammatory state and, at an early stage, the proinflammatory reaction is necessary for AT expansion. The maintenance of chronic inflammation though leads to major perturbations of AT homeostasis [8]. The triggers of inflammation are still to be better understood whereas hypertrophic adipocytes release proinflammatory factors such as TNFα, IL6 and CCL2, that promote AT macrophage infiltration [35,36,109]. The rise of the number of macrophages is not only due to the recruitment and differentiation of CCR2-dependent blood monocytes. Aouadi and collaborators demonstrated that AT macrophages (ATMs) can locally proliferate in at least a partially CCL2-dependent manner in the visceral AT of mice [110]. Functionally, obesity reduces the endocytic capability of vascular-associated ATMs [111]. Furthermore, ATMs undergo “unconventional” metabolic activation [112] rather than a classically activated (M1) or alternatively activated (M2) macrophage polarization [113]. Studies using single-cell RNA sequencing (scRNA-seq) analyses revealed that the transcriptomic profiles of ATMs are more heterogeneous in obese AT than anticipated [39,40]. A population of lipid-associated macrophages (LAMs) that accumulate at high numbers in obesity has been recently described. LAMs are characterized by the expression of the lipid sensor TREM2 that drives gene expression programs involved in phagocytosis, lipid catabolism, and energy metabolism [40]. Impairment of this process in obese mice with a global deficiency of *Trem2* inhibited the downstream molecular LAM program, leading to adipocyte hypertrophy associated with worsened metabolic condition [40]. Metabolic adaptation is a key component of macrophage plasticity and polarization [114]. Interestingly, it was demonstrated that murine macrophages acquire mitochondria from adipocytes in vivo [115]. This adipocyte-to-macrophage transfer of mitochondria defined a distinct tissue macrophage subpopulation found to be reduced in obesity. Interestingly, the inhibition of this transfer resulted in decreased energy expenditure and increased weight gain and glucose intolerance [115]. Whereas the adipocyte-to-macrophage transfer of mitochondria illustrates again the importance of cell crosstalk in AT, the mechanisms involved are still unclear. Previous studies have identified extracellular vesicles as a potential vehicle for intercellular transfer of mitochondria [116,117].

## 4. Mechanical Regulation of Adipocyte

### 4.1. Physical Constraints to and from Adipocytes

Few studies have explored the mechanical properties of adipose cells and how this may impact on their functions. Adipocytes are characterized by a unique large spherical lipid droplet which is surrounded by a thin rim of cytoplasm containing a crescent shaped nucleus and sparse organelles, mainly elongated mitochondria [118,119]. Upon intense lipolysis, smooth endoplasmic reticulum is visualized in tight contact with the lipid droplets [118,119]. During caloric excess promoting adipogenesis, the accretion of lipid droplet requires major rearrangement of the cytoskeleton [120,121] also involving changes in the mechanical properties of white adipocytes [73]. For example, adipogenesis is associated with a progressive cell stiffness increasing from 300–900 Pa in 3T3-L1 preadipocytes to 2 kPa in differentiated 3T3-L1 cells [122]. The stiffness of the lipid droplets is 2.5 to 8.3 times greater than the effective stiffness of the surrounding cytoplasm [123]. This suggests that mechanical properties of adipocytes vary according to lipid droplet size.

Moreover, it can be estimated that physical tension applied from the interior to the adipocyte membrane is generated from the lipid droplet [124]. Membrane caveolae are invaginated structures specialized in the response to mechanical tension at the cell surface membrane [125] and are particularly abundant in mature adipocytes [126,127]. It has been shown that these structures are involved in buffering lipid storage of fat cells [128]. Of interest, the importance of caveolae in the adipose cell biology is illustrated by the severe lipoatrophic syndrome that develops in patients with caveolin gene mutations [129].

Thus, in obesity, adipocyte hypertrophy is associated with increased AT stiffness [130,131,132]. Stiffness of hypertrophic adipocytes may impose mechanical constraints on the surrounding microenvironment perturbing local cell biology and eventually promoting ECM remodeling (Figure 1). On the other hand, AT fibrosis limits adipocyte hypertrophy as a biophysical constraint; e.g., adipocytes trapped in fibrotic bundles are smaller in size compared with the other adipocytes outside the bundles from the same tissue [133]. Interestingly, in a diet-induced obesity rat model, surgery-induced weight loss was related to an improvement of the biomechanical properties of visceral AT compared with “sham surgery” in obese rats. These changes in biomechanical properties were associated with enhancement of angiogenesis, reduction in adipocyte size and downregulation of the expression of ECM components [134]. Furthermore, modelling the physical constraints applied to adipocytes in ex vivo systems showed that the mechanical compression leads to major adipocyte dysfunction characterized by increased production of inflammatory adipokines as well as dysregulated lipolysis and perturbed insulin sensitivity in mature adipocytes [135].

### 4.2. Environmental Stiffness and Regulation of Adipogenesis

From a cellular perspective, a major source of knowledge comes from cell culture models involving fibroblast-like cells that differentiate to form adipocytes in response to hormonal cocktails. Observations indicate a high degree of intracellular and extracellular structural remodeling during adipogenesis [136]. Along with the biochemical properties of the progenitors’ microenvironment in vivo [137], mechanical cues impact on both stem cell proliferation and differentiation [138]. The transmission of extracellular mechanical force to the inner part of the cell, the nucleus, is achieved by a dynamic regulation of membrane and cytoskeleton integrity and tension [139]. Adipogenesis is achieved by controlling substrate mechanics, nanotopography (referring to the specific surface features generated at the nanoscale) and cell spreading [140,141]. For example, in two-dimensional (2D) cell culture models, adipogenesis requires cell confluence (e.g., cells are physically in contact with each other) [142,143]. Upon confluence, contact inhibition will arrest preadipocyte growth. In vitro adipogenesis is induced by hormonal stimulation, where preadipocytes re-enter the cell cycle, arrest proliferation and undergo terminal adipocyte differentiation [144,145]. Furthermore, three-dimensional (3D) cell culture within a methylcellulose gel promotes adipogenesis by constraining the shape of the cell similar to what is observed in confluent cells [144]. Mesenchymal stem cells plated on a small surface leading to a rounded morphology tend to undergo adipogenesis, whereas single cells grown on larger surfaces allowing cell spreading tend to undergo osteogenesis [136,141,146,147,148]. As cellular shape is influenced by mechanical properties and forces exerted on the cell, it is thus not surprising that mesenchymal stem cell fate is determined by substrate mechanical cues [138]. In mesenchymal stem cells, adipogenesis is favored by a soft microenvironment [140,149]. Thus, culturing ASCs on gels that mimic the native stiffness of AT (2 kPa) encourages adipogenesis, even without exogenous adipogenic hormonal stimuli [150]. When human mesenchymal stem cells are seeded on polyacrylamide gels with low stiffness, they are more prone to become adipocytes than cells grown on stiffer gels [150]. When microenvironment stiffness increases, the ASC spread out and lose their rounded morphology. Moreover, cyclic stretching or vibration impedes adipocyte differentiation, but static stretching promotes it [122,123,151]. Altogether, cell spreading and mechanical cues can constrain adipogenesis by modifying cell shape [136].

### 4.3. Influence of Extracellular Matrix and Osmotic Microenvironment on Adipose Cells

ECM is a major determinant of AT mechanical properties. Knockout of genes that encode collagens [66] or collagenases induces modulation of pericellular collagen rigidity and affects adipogenesis [152]. Dani and collaborators demonstrated that the inhibition of collagen synthesis with a competitive inhibitor of prolyl hydroxylase (Ethyl 3,4-dihydroxybenzoate) represses the differentiation of preadipocytes into adipocytes and impairs ECM development [153]. In vitro, type I collagen has been reported to favor adipogenesis [154], and denatured, but not native, type IV collagen, promotes adipogenesis in human mesenchymal cells [155].

Microenvironment stiffness and composition are thus under the control of the surrounding ECM that also involves members of the matrix metalloproteinase (MMP) family, and their functions are coordinated by inhibitors of metalloproteinases called TIMPs (for tissue inhibitors of metalloproteinases). Metalloproteases (MMP), where their expression in AT and serum is increased in obesity [156,157,158], exert effects on adipogenesis. For example, MMP14 has a major effect on adipogenesis by altering collagen stiffness in a 3D setting in vitro [152]. *Mmp14*^+/−^ mice challenged with HFD have defects in AT expansion due to impaired type I collagen cleavage, and develop metabolic alterations [159].

Another ECM protein, fibronectin, inhibits the differentiation of 3T3-F442A preadipocytes into adipocytes [160]. Similarly, the presence of secreted-protein-acidic-and-rich-in-cysteine inhibits adipogenesis by stimulating the deposition of fibronectin [161]. The anti-adipogenic activity of fibronectin can be reversed by keeping cells in a rounded configuration or by exposing cells to cytochalasin D, which disrupts the actin cytoskeleton [160].

Blocking the interaction of integrin alpha 6 with laminin by an anti-integrin alpha 6 antibody, enhances differentiation of adipocytes by repressing RhoA (characterized by anti-adipogenic effects) [162]. Proper integrin signaling is also required for de novo beige fat biogenesis following cold exposure. The integrins β5 and β1 interact with CD81, and are involved in irisin-mediated FAK signaling in beige ASCs [163]. Knockout or antibody-based blockage of either integrins (β5 or β1) or CD81 abolishes the effect of irisin-induced FAK phosphorylation in beige ASCs [163]. Noteworthy, the number of CD81 and ASCs negatively correlates with body weight, insulin resistance, glucose intolerance, and WAT inflammation in experimental models of diet-induced obesity [163].

Another important element of the cellular microenvironment is osmotic pressure which can influence cellular membrane tension, and mechanotransduction. All cells may be exposed to osmotic swelling or shrinkage but in the case of adipocytes, the effect of osmotic pressure might be more pronounced due to the relatively small water compartment [164,165]. Interestingly, the development of the mechanisms controlling osmotic pressure is an important part of the adipocyte phenotype. As shown by Eduardsen et al., osmotic swelling promotes nuclear ERK1/2 activity and decreases IRS phosphorylation in response to insulin [166]. Alternatively, osmotic shrinkage increases phosphorylation of IRS and FAK, lowering the ERK signaling at the same time [166,167,168].

### 4.4. Effect of Mechanical Cues on Brown Cells

Brown/beige adipocytes differ from white adipocytes in their morphology and their functions. Thermogenic adipocytes need to maintain an extensive and dynamic cytoskeleton to support and organize multilocular lipid droplets and a dense network of mitochondria. This also implies significant differences in mechanical properties. BAT is stiffer than WAT, and BAT stiffness increases when mice are exposed to lower temperature (4 °C). By using atomic force microscopy, Tharp et al. observed that in vitro treatment of brown adipocytes with β-adrenergic agonist (isoproterenol) induces a contractile-like response which stiffens the cell cortex [169]. The contractile-like response in BAT appears to strongly mimic that of cardiomyocytes which can be explained by the common embryonic origin (Myf5^+^ cells) shared by brown adipocyte and muscle cells [170]. As such, muscle-specific type II myosin heavy chains (MyH) are expressed in brown fat cells conferring a dynamic role of actinomyosin mechanics in thermogenic cells [169]. The repression of actinomyosin-mediated tension by using type II myosin inhibitors (blebbistatin and 2,3-butanedione monoxime) reduces cytoplasmic stiffness and leads to a specific loss of UCP1 expression and oxidative capacity. The authors showed that actinomyosin contractility regulates the thermogenic capacity of brown adipocytes via the mechanosensitive transcriptional co-activators YAP and TAZ (see below). Indeed, YAP overexpression increases UCP1 levels. Accordingly, mice with brown adipocyte-specific (*Ucp1*-*cre*) knockout of YAP/TAZ resulted in whitening of BAT, increased fat mass, decreased energy expenditure, and showed hyperinsulinemia and glucose intolerance [169].

## 5. Actors of the Mechanotransduction: From Tension to Adipose Cell Function

Cell shape is linked to extracellular mechanical properties such as ECM stiffness, stretching or shear stresses exerted by flowing liquids and attachment to other cells (Figure 1 and Appendix D). In the case of fibrotic AT, the changes of ECM stiffness are, therefore, critical in adipocyte mechano-responses. In a model of type II diabetes mellitus, by mixing human decellularized AT ECM and adipocytes, it has been shown that the specific dysfunctional fibrotic cues are part of the ECM and not a property of cells [135]. Another research supported this finding by demonstrating that mechanical changes of the surrounding ECM are transmitted by the cytoskeletal actomyosin stress fibers in adipocytes that were cultured on stiffness-controlled hydrogels [171].

### 5.1. Cytoskeleton

The cytoskeleton is a dynamic structure essential to maintain cell shape, which provides structural support that functions as a mechano-sensor in combination with focal adhesion proteins [172]. During adipocyte differentiation, ASC undergo morphological transformation to allow the formation and growth of nascent lipid droplets [173] where vimentin transcription increases to proportionally expand the vimentin cage around each droplet [174], while tubulin and actin expression decreases [121]. These cytoskeletal rearrangements are prerequisites for terminal differentiation [121,160,175,176,177]. The inhibition of actin filament polymerization or actomyosin contraction promotes a rounded morphology of adipocytes and subsequently promotes adipogenesis [141,178].

During adipogenesis, changes in cell shape involve disruption of filamentous (F) actin via downregulation of RHO GTPase–RHO-associated kinase (ROCK) signaling and the actin-regulatory proteins (Arp2/3 complex, cofilin, profilin) [120,141,179]. The inactive RHO, RHO GDP, is the predominant RHO species in confluent or rounded human MSCs and promotes adipogenesis, whereas ectopic addition of RHO GTP inhibits it [141]. However, in stiff environments, focal adhesion assembly is promoted, RHO GTP activates ROCK, which, in turn, stimulates F-actin stress fibers formation and translocation of YAP/TAZ to the nucleus, inhibiting adipogenesis [141,180] (Figure 2).

In cultured mature adipocytes, lipid droplet growth is associated with increased Rho-kinase activity [181]. Accordingly, mechanical stretching (e.g., mechanical event induced by lipid droplet enlargement in the cell) on mature adipocytes increases Rho-kinase activity and stress fiber formation, as observed in adipocytes from HFD-fed mice [181]. Thus, adipocyte hypertrophy following 2 weeks of HFD-feeding in mice was associated with a drastic increase in F-actin, increased Rho-kinase activity, and changed expression of actin-regulating proteins, favoring polymerization [182]. Conversely, another study reported that lipid droplet enlargement in adipocytes downregulates cortical F-actin formation [183]. Interestingly, the authors proposed that this downregulation of F-actin organization would lead to decreased insulin-dependent GLUT4 trafficking to the plasma membrane, which governs the insulin-dependent glucose utilization [183].

### 5.2. Mechano-Induced Cell Metabolism Drives Cell Functions

Recent advances have revealed that the activation of transcriptional programs that determine cell functions can be modulated by the biochemical environment shaped by cell intrinsic metabolic activity exhibited by a cell. Indeed, metabolism impacts intracellular cell signaling pathways by influencing protein post-translational modifications (PTMs) of signaling mediators. Moreover, these PTMs impart a broad influence on transcriptional activation by modulating chromatin state via DNA (hydroxy)methylation, histone or transcription factors PTMs [184,185,186,187,188]. Thus, altered metabolism has been proposed to play an important role in AT dysfunction altering, in turn, whole body energy homeostasis [5,189,190,191].

Actually, cell metabolism is tightly regulated by mechanical cues. Recent work revealed that mechanical forces impact cell metabolism via cytoskeletal reorganization [192,193]. Park et al. found that increasing substrate stiffness correlates with the trapping of E3 ubiquitin ligase tripartite motif (TRIM)-containing protein 21 (TRIM21) within actin stress fibers, thus reducing access to its substrates such as phosphofructokinase (PFK) [194]. On a soft ECM that permits relaxation of the actomyosin cytoskeleton, TRIM21 is released, PFK is degraded thus leading to reduced glycolysis [194] (Figure 3). Accordingly, matrix stiffness increases ATP:ADP ratio and glucose uptake by stimulating glucose transporter 1 (GLUT1) [195,196]. GLUT1 has been reported to be retained at the site of force transmission by ankyrin G which is critical for allowing cells under tension to tune glucose uptake and to fuel the reinforcement of the actin cytoskeleton [196,197]. Moreover, YAP and TAZ that relay stiffness signals, stimulate glucose metabolism by enhancing GLUT1 expression [197]. Bertero et al. also reported that glutamine catabolism is dependent on YAP/TAZ that bind to the promoter of GLS (glutaminase) to promote its transcription [198,199]. Thus mechano-activation of glutaminolysis and glycolysis sustains the metabolic needs of mechano-activated cells and, in turn, alters cell phenotype. In order to elucidate the outmost importance of the mechanics of the environment on adipocyte behavior and the impact of fibrosis on their identity, traditional in vitro models have to be replaced by higher-fidelity microphysiological systems that can mimic physiopathological conditions, as reviewed recently in [200].

### 5.3. The Nucleus

The nucleus is the largest and stiffest cellular organelle [172]. In adipocytes, the nucleus is compressed against the plasma membrane by large lipid droplets. It can be divided structurally and functionally into the nuclear interior, which houses chromatin, and the surrounding nuclear envelope (Figure 1). The nuclear envelope is composed of two concentric membranes, e.g., the inner and outer nuclear membranes, an underlying nuclear lamina, and nuclear pore complexes that control entry of large molecules into the nuclear interior [172]. The nuclear lamina is a filamentous protein network lining the nucleoplasmic surface of the inner nuclear membrane. It is composed of A-type and B-type lamins, and lamin binding proteins [172,201]. Although B-type lamins are mostly restricted to the nuclear lamina, A-type lamins (encoded by *LMNA*) are found both at the nuclear periphery and in the nuclear interior interacting with chromatin [202,203,204,205]. When A-type lamin is depleted, cellular elasticity and viscosity of the cytoplasm decrease and the nucleus becomes softer and more deformable [206]. On the other hand, seeding adipocytes on stiff surfaces and subsequent spreading of cells is associated with upregulation of lamin A proteins [207].

During cell differentiation, the nuclear architecture is reorganized [208,209]. For example, in preadipocytes, A-type lamins move from intranuclear structures to the nuclear rim as adipogenesis proceeds [209]. The important role of lamin is illustrated by mutations throughout the *LMNA* gene that cause various forms of laminopathies, including partial lipodystrophies (Appendix E) [210].

Lamins play an important role in physically connecting the nucleus to the cytoskeleton. Such connections are mediated by the members of the so-called linker of nucleoskeleton and cytoskeleton (LINC) complex [211,212]. These connections mediate intracellular force transmission, cell migration and cell polarization [213]. Through these connections, when mechanical force is applied on the plasma membrane, alterations of the nuclear morphology are induced [214] and translated into chromatin remodeling and epigenetic modifications [215,216]. Interestingly, by using photoconvertible, photo-softening [215] and photo-stiffening [217] hydrogels, Anseth and collaborators demonstrated that human mesenchymal stem cells grown on stiff matrices undergo chromatin remodeling that is characterized by increased histone acetylation and reduced chromatin condensation. Then, an interesting model to consider would be that mechanical cues from the microenvironment are transmitted through integrins, the cytoskeleton, the LINC complex, and lamins to the nucleus, where the mechanical cues are translated into chromatin remodeling and epigenetic modifications [218,219]. Reports have shown how mechanical cues can regulate epigenetic status, through the influence of actin cytoskeleton [216,220,221]. It has been shown that alteration in cytoskeleton tension by compression/stretch in the direction perpendicular to cell alignment induces nuclear shape change, a decrease in histone deacetylase (HDAC) activity, and an increase in histone acetylation [222]. In addition, cells in an elongated shape exhibit elongation of their nucleus and higher levels of nuclear histone H3 acetylation, compared with cells in a circular shape [223]. Although not in adipose cells, this discovery demonstrates the impact of mechanical cues to influence the epigenetic state [223]. The mechanism behind this “biophysical” alteration of histone acetylation relies on altered HDAC activity, possibly because HDAC is sequestered in the cytosol by actin cytoskeleton [224]. Thus, the dynamic reorganization of the nuclear lamina and the cytoskeleton, induced by mechanical cues, may impact on histone acetylation that plays a key role in controlling adipogenesis and adipocyte function [225].

### 5.4. Mechano-Transduction Mediated by YAP/TAZ

Yes-associated protein (YAP) and PDZ-binding motif (TAZ) are paralogous proteins that act as transcriptional co-regulators [226,227]. The phosphorylation of YAP/TAZ by LATS1/2 (Hippo signaling) leads to cytoplasmic retention and degradation of YAP/TAZ [228]. YAP/TAZ are robust mechano-sensors that mediate cell responses to cell spreading [229,230] and stretching [231], fluid flow-induced shear [232], and to microenvironment stiffness [180,226]. However, the mechanisms by which mechanical cues regulate YAP/TAZ nuclear shuttling are still unclear.

During adipogenesis, activation of LATS2 and concomitant reduction in YAP/TAZ nuclear activity were described [233]. The cytoplasmic retention of TAZ acts on Wnt signaling to prevent β-catenin translocation to the nucleus, which stimulates the pro-proliferator and anti-adipogenic TCF/LEF transcriptional activity [233]. Interestingly, osteoblast-specific deletion of TAZ in mice reduces cell proliferation and osteogenesis, and promotes adipocyte formation, resulting in a trabecular bone loss. Conversely, YAP is selectively expressed in osteoblast-lineage cells where it interacts with β-catenin, increasing β-catenin stability in the nucleus [234]. Interestingly, YAP repression by phosphorylation appears to be mediated by cell spreading and is sufficient to prompt adipogenesis, regardless of a permissive environment [149]. However, ASC-specific overexpression of YAP appears to increase adipogenesis and obesity in mice [235]. Nevertheless, this study reported that Yap overexpression in ASC induced a negative feedback mechanism on the Hippo signaling pathway, leading to suppression of TAZ activity. The latter enhances PPARγ activation and increases adipogenesis [235]. It was also proposed that increased cytoskeleton formation, by overexpression of smooth muscle actin (SMA), induces cell spreading, increases YAP activity and inhibits adipogenesis in human mesenchymal stem cells [236]. Together, these studies suggest that the YAP/TAZ pathway mediates adipogenic regulation induced by microenvironment stiffness and cell spreading.

As adipocyte hypertrophy increases local mechanical stress, it was observed that YAP and TAZ were both activated in mouse and human white adipocytes during obesity [237]. In agreement with the anti-adipogenic effect of YAP/TAZ activity, Olefsky and collaborators have shown that TAZ acts as a PPARγ co-repressor in adipocytes. In obese mice, adipocyte-specific deletion of TAZ increased PPARγ activity and resulted in increased systemic insulin sensitivity and glucose tolerance with an improved AT phenotype [238]. However, compelling and contradictory data demonstrated that the loss of both YAP and TAZ specifically in adipocytes resulted in lipodystrophy, including massive adipocyte death, increased adipogenesis, macrophage infiltration and improved glucose tolerance in HFD fed mice [237]. There are several hypotheses to reconcile these paradoxical results. One possibility is that YAP and TAZ work redundantly to maintain adipocyte survival so that the loss of both simultaneously is fatal for adipocytes. The second possibility is that TAZ has an activity not shared with YAP. TAZ, where its role is better described, represses PPARγ activity in white AT that contributes to insulin resistance, abnormal AT remodeling and local inflammation [238,239,240]. However, the role of YAP in regulating adipocyte differentiation or adipocyte phenotype appears uncertain. It could be involved in the negative feedback mechanism via the Hippo signaling pathway or in the control of cell death [237].

### 5.5. Membrane

The conversion of forces can also be transduced through electrical signals via the opening of mechanosensitive ion channels present in the cellular membranes [241]. In addition to the above-mentioned regulation of caveolae assembly/disassembly, increased adipocyte volume and subsequent increase in plasma membrane tension may result in the activation of these ion channels. Recently, evidence was provided that two different channels, Piezo1 and SWELL1 (a volume-regulated anion channel (VRAC)), are activated in the context of adipocyte hypertrophy [242,243]. It has been shown that SWELL1 activation in response to mature adipocyte volumetric expansion promotes adipocyte expansion, energy storage, and enhances insulin signaling during increased caloric intake [242]. Piezo1 in mature adipocytes is rather a critical mediator of obesity-induced adipocyte hyperplasia. Indeed, Offermanns and collaborators have shown that the opening of Piezo1 in hypertrophic adipocytes induces the release of the pro-adipogenic fibroblast growth factor 1 (FGF1) which promotes, in turn, ASC differentiation [243]. These data demonstrated that the mechanosensitive ion channels are cell-autonomous sensors of adipocyte volume that regulate AT growth, insulin sensitivity and adipogenesis (Figure 4).

## 6. Conclusions

The current obesity epidemic needs a great deal of attention on the understanding of adipose biology. Mechanical properties are spatially and temporally regulated to preserve the homoeostasis of AT. Indeed, the adipose microenvironment is a mechanically complex niche, in which a number of inputs regulate AT function. However, the influence of mechanical cues on adipose function, and vice versa, remain underappreciated. The interplay between adipose resident cells and ECM determines the maintenance of a mechanical environment that supports tissue architecture and function. Aberrations in adipose microenvironment play a profound role in the development and progression of obesity. Investigations have partially revealed how adipose cells respond to mechanical stress during AT expansion. There is also evidence that a niche can modulate preadipocyte and adipocyte fate by defining the mechanical properties. We believe that fibrosis and adipocyte hypertrophy ultimately disrupt mechanical homeostasis, altering mechanotransduction signaling through cell–cell, cell–matrix–mediated transcriptional/epigenetic and PTMs mechanisms, leading to progressive ECM deposition and AT dysfunction. Thus, it will be important to dissect the crucial molecular events that connect mechanical cues with defects in AT function. Ultimately, translating these insights into clinical and therapeutic interventions may enable to delay the cycle of obesity and its related complications.

## Figures and Tables

**Figure 1 cells-11-02310-f001:**
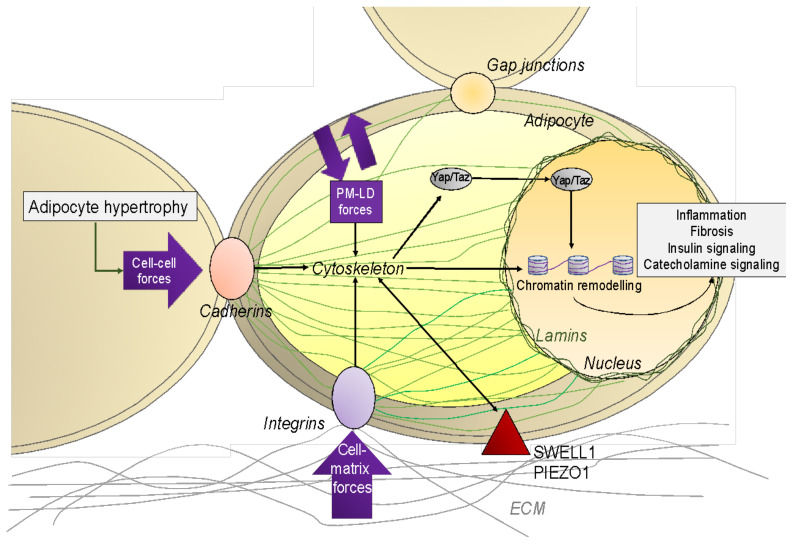
**Mechanosensing mechanisms in adipose tissue.** Adipocytes receive mechanical forces at the cell plasma membrane arising from the surrounding extracellular matrix (ECM) or from neighboring cells via integrins, cadherins, GAP junctions or other mechanosensitive proteins such as SWELL1 or PIEZO channels. Mechanical cues can emanate from cell–cell adhesion, cell–ECM adhesion, cytoskeleton remodeling (actomyosin, stress fiber, etc.) and from the growth of lipid droplet (plasma membrane (PM)–lipid droplet (LD) forces) that stiffen the cells (purple arrows). Force sensing and transmission at cell–ECM and cell–cell adhesions converge on the cytoskeleton and can directly be transmitted to the nucleus, resulting in chromatin remodeling and transcriptional changes to modulate cell phenotype (inflammation, fibrosis, insulin resistance, and catecholamine resistance).

**Figure 2 cells-11-02310-f002:**
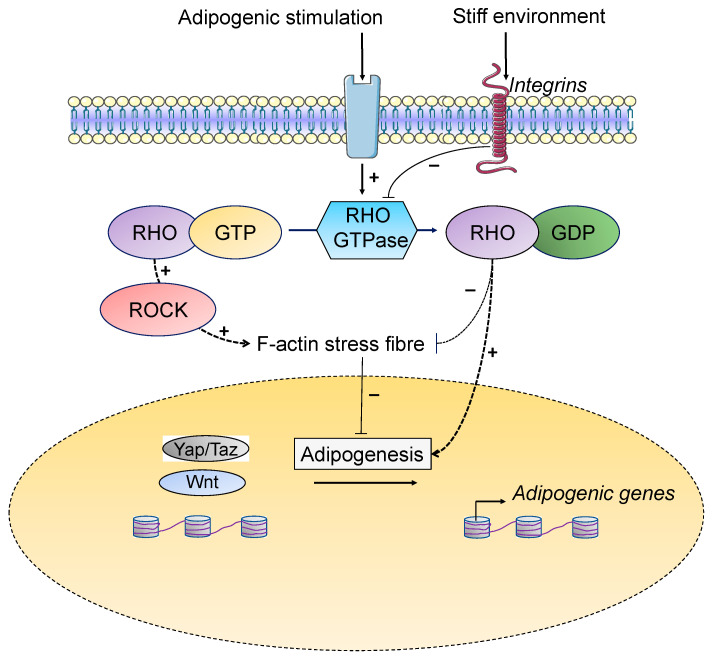
**Microenvironment stiffness can regulate adipogenesis.** During adipogenesis, the cells undergo substantial morphological modifications where spindle-shaped fibroblast-like preadipocytes become rounded mature adipocytes. There is an extensive reorganization of the cytoskeleton that allows lipid droplet growth and expansion. Upon adipogenic stimulation, the inhibition of Ras homolog family member A (RhoA) and Rho-associated protein kinases (ROCKs) disrupts actin cytoskeleton structures allowing adipogenesis. Thus, the inactive form, RHO GDP, promotes adipogenesis. However, in stiff environments, focal adhesion assembly is promoted, RHO GTP activates ROCK, which, in turn, activates F-actin stress fiber formation and translocation of YAP/TAZ to the nucleus, which breaks adipogenesis.

**Figure 3 cells-11-02310-f003:**
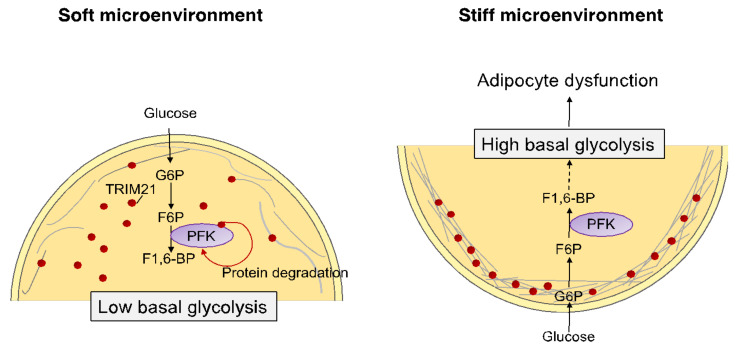
**Hypothetical mechanical regulation of metabolism.** Recent studies revealed that mechanical forces altered cell metabolism via cytoskeletal reorganization in cancer cells [194]. In a soft microenvironment, TRIM21 was not trapped by actin stress fiber bundles, which degrade phosphofructokinase (PFK). This led to low glycolysis rates. By contrast, when cells were surrounded by a stiff microenvironment (as in obese adipose tissue), TRIM21 was bound to the F-actin bundles of the cytoskeleton and thus PFK degradation was prevented, leading to high rates of glycolysis. By these mechanisms, we believe that mechanical cues could influence the metabolic phenotype and progression of adipose tissue dysfunction.

**Figure 4 cells-11-02310-f004:**
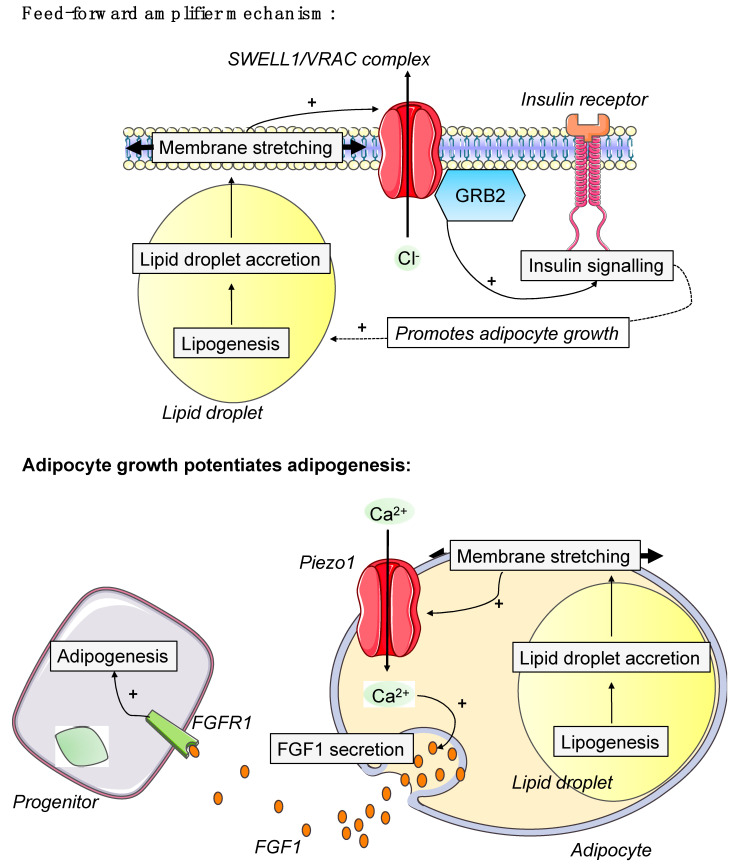
**Roles of mechanosensitive channel in adipocyte.** SWELL1/VRAC complex or Piezo1 channels are activated in response to increases in adipocyte volume. SWELL1/VRAC activation stimulates insulin–PI3K–AKT2 via GRB2, and thereby supports lipogenesis and continued adipocyte growth in a feed-forward manner [242] However, opening of Piezo1 in mature adipocytes causes the release of the adipogenic fibroblast growth factor 1 (FGF1), which induces adipogenesis through activation of the FGF-receptor-1 [243].

## Data Availability

Not applicable.

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
