# Peer review of "Importance of the Microenvironment and Mechanosensing in Adipose Tissue Biology"

_cells, 2022, doi:10.3390/cells11152310_

Round 1

Reviewer 1 Report

This manuscript reviews the importance of the microenvironment and mechanosensing in adipose tissue biology in both health and disease, especially obesity. In my opinion, both well-known alterations (such as hypertrophy vs. hyperplasia and Inflammation and fibrosis) and more novel issues (such as connexins and neuronal fibers and physical constraints to and from adipocytes) are adequately explained. The drawings seem to me to be accurate and clear. I do not quite understand the reason for appendices A-E, which I think could be perfectly included in the text.

Finally, I miss some previous reviews on the subject, for example: Unamuno et al. “Adipokine dysregulation and adipose tissue inflammation in human obesity”, Eur J Clin Invest. 2018 Sep;48(9):e12997. doi: 10.1111/eci.12997. I do not pretend to say that it has to be cited but I would want to know how the selection has been made.

Author Response

The appendices that provide basic information on adipose tissue biology were considered as minor by the reviewer #3 and recommended to remove them. Indeed, our aim with them is to make available basic information on adipose tissue biology for the non-specialists (like mechano-biologists). To moderate both opinions, we opt to maintain them as they are in this review.

We have selected the most relevant references that support our aim in this review and we might have missed some others. We have now added the reference suggested by the reviewer.

Reviewer 2 Report

This study comprehensively describes the physical features and mechanical regulation of adipose tissue. This manuscript is well-written and provides very educative and sufficient findings. The reviewer’s concerns only include the editorial or minor points. For example, in p13 lines 549-550, the same sentences are duplicated. In line 551, the font of PPARγ is strange. In addition to these, there are probably such errors. Please check the whole manuscript.

Author Response

We have carefully revised the manuscript to ensure that the text is quite clear and free from typographical and grammatical errors.

Reviewer 3 Report

The work presented by Lecoutre and colleagues describes a process that is fundamental in adipocyte biology, mechanotransduction, which little is poorly known. In this line, the review has high interest to the scientific community, however there are several (minor) points that could be improved for publication:

-       Based on the objective described in the review focused on role of mechanosensitive pathways in regulating AT functions the first paragraph are less novel and could be reduced and extend the knowledge described in adipose tissue related to ECM remodelling. It should be interesting include additional information and novel advances related to fibrosis in adipose tissue. In this line, there are several articles that has really interest in the adipose tissue fibrosis and obesity with are not mentioned.Acta Biomater. 2022 Mar 15;141:264-279.  doi: 10.1016/j.actbio.2022.01.007.  Epub 2022 Jan 8.; FASEB J. 2020 Jun;34(6):7520-7539. doi: 10.1096/fj.201902703R. Epub 2020 Apr 15.; Mol Metab. 2019 Jan;19:97-106. doi: 10.1016/j.molmet.2018.10.007. Epub 2018 Oct 23, due to not only collagen or nidogen are increase in obesity (lines 115-117).

-       Several appendix are too extensive and include additional information that is known, and probably could be also reduced. 

Author Response

We have taken the comments on board to improve the manuscript. Most of the novelties coming from the articles suggested by the reviewer are now discussed in the manuscript.

The appendices that provide basic information on adipose tissue biology were considered essential by the reviewer #1. Indeed, our aim with them is to make available basic information on adipose tissue biology for the non-specialists (like mechano-biologists). To moderate both opinions, we opt to maintain them as they are in this review.